# The Effect of Mixing Pressure in a High-Pressure Machine on Morphological and Physical Properties of Free-Rising Rigid Polyurethane Foams—A Case Study

**DOI:** 10.3390/ma16020857

**Published:** 2023-01-16

**Authors:** Grzegorz Węgrzyk, Dominik Grzęda, Joanna Ryszkowska

**Affiliations:** Faculty of Materials Science and Engineering, Warsaw University of Technology, ul. Wołoska 141, 02-507 Warsaw, Poland

**Keywords:** polyurethane foam, high-pressure machine, process parameters, pouring parameters, polyurethane mixing

## Abstract

This article presents the results of testing foam blocks made with a high-pressure foaming machine under industrial conditions. Foam blocks were made at pressures in the range of 110–170 bar with substrate temperatures allowed by machine suppliers. The foaming process parameters of each block were evaluated. The structure of the foams in the outer and central parts of the blocks was characterized using FTIR spectroscopic analysis and microscopic observations using SEM. The changes in apparent density, strength properties and brittleness of the foams were evaluated. The properties of the blocks made at different mixing pressures varied depending on the pressure at which the substrates were mixed and the location in the block. The biggest differences that were observed were the friability of the foams taken from different locations in the blocks by up to about 30%; the apparent density differed by about 8% and the compressive strength by about 5%.

## 1. Introduction

Polyurethane foam, the material invented by Otto Bayer, revolutionized the entire world, and from the very beginning in 1987, almost 2.7 billion pounds were produced and sold [1]. The global polyurethane industry market in 2020 was valued at almost 71 billion USD and is expected to rise at a compound annual growth rate (CAGR) of 3.8% from 2021 to 2028 [2]. This tremendous success is mainly related to the increasing demand for building insulation due to the fact that polyurethane foams and polyisocyanurate foams provide one of the best insulation properties [3]. Its implementation in the early stages of construction may save energy loss and positively influence the environment and costs. Polyurethane materials and composites are widely applied to many fields of industry, not only fields such as automotive [4,5], furniture, and interiors [6], but also medicine [7,8,9].

In order to steadily increase demand for polyurethane foams, it is necessary to produce foams of the most reproducible and high quality possible, which depends, among other things, on the foaming process. Improvement for this application is a clue, taking into account the fact that market demands are rising.

A polyurethane foaming process consists of many stages. During it, chemical and physical processes coincide. As a function of time, the group of physical processes are those associated with mixing and subsequent entrainment of bubbles (bubble entrainment), increase in viscosity, increase in modulus, cracking of cells, and achievement of final characteristics of foams. In the group of chemical processes are kinetic processes and those affecting the morphology of foams. The kinetic processes are, in turn, the reaction of water with NCO groups leading to the release of CO_2_, the reaction of a polyol with NCO groups leading to an increase in the molecular weight of the resulting polymer, and the formation of cell networks (cell reticulation) and foaming (blow off). Phenomena affecting the morphology of foams are, in turn, the formation of bubbles (bubble formation), the process of phase separation, and the opening of pore windows resulting from the formation of ureas [10]. The course of bubble entrainment is linked to CO_2_ generation and bubble growth [11]. In the paper of Kanner and Decker [12], it was shown that after the mixing stage associated with bubble entrainment, no new bubbles are formed, and the subsequent processes are bubble growth and coalescence. These observations were confirmed by a study conducted by Brondi and co-authors [13]. As a result, they found that when a large number of air bubbles are introduced during rapid mixing, no further nucleation of air bubbles occurs. In contrast, during slow stirring, no air bubbles are formed, but any bubbles are the result of nucleation from CO_2_ and the introduction of pentane. The work of Kanner and Brondi concluded that for the foaming process, the work put into mixing is an essential element in order to obtain an optimal polyurethane product.

There are three basic chemical methods of synthesis: one step (one shot), quasi-pre-polymer, and full pre-polymer [14]. The most common method is the one-step method because of its easy implementation into production processes. The one-step method can be easily implemented into productions by applying a mechanical stirrer, which is often used in the technique. This is also often the mixing method used in laboratories and research described in the literature [13,15,16]. When it comes to industrial applications, mixing by hand is subjected to error, low speed of production and difficulties in automatization, mainly when the production line is not operated by chemists. The single-stage production of polyurethanes can also be carried out thanks to the use of an appropriate machine park, which ensures better mixing, dosing stability and repeatability of quality, provided that the processing parameters are correctly selected. The use of mixers results in fewer production defects and lower production costs of products.

Mixing machines allow the introduction of a large amount of energy in the foaming process, on which not only the process depends, but also the quality of mixing the ingredients [17].

One type of machine used to make rigid polyurethane foams (RPURFs) is a high-pressure mixing machine. It is a machine often used in production processes based on impact mixing. This is a mixing process in which two opposing streams meet at high speed so that they are mixed together. One of the basic parameters for manipulating the energy of opposing jets is the mixing pressure, which, according to various machine manufacturers, should be between 100 and 200 bar [18,19,20].

Therefore, in this article, the main emphasis was placed on the influence of the mixing pressure on the quality of the obtained products. Our article examines open-die free block casting to simulate the basic manufacturing process of parts that are obtained from the block-cutting process. In this type of production, the most important factor is the even distribution of foam properties throughout the block to avoid fluctuations in parameters in the products to be cut.

## 2. Materials and Methods

### 2.1. Materials

#### Synthesis Process

A set of rigid polyurethane foam blocks were manufactured using a high-pressure industrial machine brand of EKOSYSTEM SRL, model Hydra 100. An example of a scheme of a high-pressure machine is presented in Figure 1. A sample of water-based polyurethane systems—rigid polyurethane foams (RPUFs), was obtained from the company that also shared the machine for this production. The name and recipe of the mixture remain the know-how of the company that cannot be shared.

The exact mass of the manufactured block was 600 g, and the pouring time was 1 s, providing a total flow of 600 g/s. According to the datasheet of the polyurethane provider, the ratio for the system was *r* = 1.4:(1)$$r=\frac{Q_{ISO}}{Q_{POL}}$$
where *Q* is a flow of each component [g/s].

The machine was prepared to work with pressures between 100–180 bars, which is why the pressures chosen for this experiment were between 110, 130, 150 and 170 bars. It is not possible to work at extreme values because of occurring alarms.

### 2.2. Sampling and Nomenclature

A piece of foam in an open mold was manufactured in the shape of a rigid foam block (Figure 2a). After two weeks of seasoning at room temperature, blocks were cut into smaller blocks, as shown in Figure 2b. Then this rectangular block was cut into samples: A from the edge along the shorter side of the cuboid, and sample B as its remaining middle part, as shown in the diagram in Figure 2b.

The test specimens were determined using the mixing pressure and the site from which the specimens were cut, e.g., 110A means a sample from the block manufactured with mixing pressure 110 bars and localization A (a side of the block) from this block.

### 2.3. Synthesis Parameters

An electronic stopwatch with an accuracy of 1 s was used to determine a take-off time (the time elapsed from the combination of components A and B to the start of foam build-up), a build-up time (to the maximum foam height), and a gel time (until the mixture’s viscosity is sufficient to pull a strand out of the polymer with a rod), and the tack-free time (when a surface of foam is touched, it does not stick).

### 2.4. Apparent Density

The apparent density was calculated by measuring the weight and volume of the sample according to EN ISO 845. The weight of the samples was determined with an accuracy of ±0.1 mg using a WPA 180/C/1 analytical balance (Radwag, Poland), and 50 × 50 × 25 mm cubes cut from the samples were measured with an accuracy of ±0.1 mm.

### 2.5. Fourier Transform Infrared Spectroscopy (FTIR)

The chemical composition of the foams was analyzed using absorption spectra obtained with a Nicolet 6700 spectrophotometer (Thermo Electron Corporation, Waltham, MA, USA) equipped with an attenuated total reflection (ATR) module. Each sample was scanned 64 times in the wavelength range of 4000–400 cm^−1^. The results were analyzed with Omnic Spectra 8.2.0 software (Thermo Fisher Scientific Inc., Waltham, MA, USA).

### 2.6. Scanning Electron Microscopy (SEM)

Samples were observed using a Hitachi TM3000 (Hitachi High-technology Corporation, Tokyo, Japan). Before observation, the samples were sprayed with a palladium–gold layer. Imaging was performed with secondary electrons at an acceleration voltage of 5 kV. Porosity was assessed in pore size, shape and spatial distribution based on several images taken at ×40 magnification. SEM images were used to calculate the mean equivalent diameter and anisotropy index of pores (N ≥ 100 for each foam variant). The anisotropy index was calculated as the ratio of the cell height to width, later called the aspect ratio. Image analysis was made using ImageJ software, which allowed the measurement of the parameters mentioned above.

### 2.7. Friability

The friability test was performed according to ASTM C421-08.

### 2.8. Determination of Compression Properties

The compression test was performed according to the standard ISO 844 in both directions—perpendicular and parallel to the direction of growth.

### 2.9. Closed Cell Content

The amount of closed cell content was measured according to the PN-EN ISO 4590-2016. Laboratory tests were performed using a self-made circuit, the same as described in method 2a in ISO standard.

### 2.10. Viscosity Test

Viscosity tests for the polyol blend were performed using a Brookfield DV-II + Pro Viscometer.

### 2.11. Water Absorption

The water absorption analysis, as well as dimensional stability, were conducted at 40 °C. For each material, three samples were weighed, followed by immersion in water for 24 h. After taking the samples out of the water and letting the water drain from the external sample surface for about 10 min, the samples were weighed again.

## 3. Results

This paper analyzes the influence of the pressure of mixing of substrate streams in a high-pressure machine on the properties of the obtained products formed at a pressure in the range of 110–170 bar. The remaining synthesis parameters are summarized in Table 1. The temperature of the polyol component was 23.5 ± 0.4 °C, and for the isocyanate component it was 23.1 ± 0.5 °C. Machine suppliers suggest using temperatures within 1 °C of difference, and this experiment was carried out in this temperature regime.

### 3.1. Synthesis Parameters

Characteristic times that describe the speed of reaction of synthesis are presented in Table 2. The start time is the time in which foams changed translucency and started rising. This is often correlated with start time—the time when the foam starts to rise. Rise time is the time when foam stops rising. The gel time is specified when a long string of polymer can be extracted during the surface touch with, e.g., a glass rod. When the polymer thread does not adhere to the rod, we can mark this as the tack-free time. Tack-free time is the time when touching the surface of the foam does not stick. For every block manufactured with different mixing pressure, the start time was higher for blocks manufactured with 110 and 130 bars, compared to blocks manufactured with higher pressure, where the start time was lower, and its value was 7–8 s. For the block prepared with a mixing pressure of 170 bar, the rising time was more or less 25 s (15%) lower than the rising time of other blocks, and the gelling time was lower by about 30 s (15%).

### 3.2. Effect of Mixing Pressure

The main cause of differences in foam properties made of the same material is the process parameters changing. A set of SEM photos of foam 110 and 170 in parallel and perpendicular view are collected and shown in Figure 3 and Figure 4.

It was observed that foams made at 170 bar contained pores of varying sizes in a wider range than in other foams.

The differences in the cross-sectional images obtained perpendicular to the direction of foam growth and the parallel direction (Figure 3 and Figure 4) are challenging to analyze qualitatively, so a quantitative analysis was performed (Figure 5).

The average pore size on cross-sections made transverse to the direction of foam growth increases with increasing mixing pressure (Figure 5). For samples taken from site A and for B, the increase in pore size is small. Analysis of cross-sections made parallel to the direction of foam growth reveals that, for all sites sampled, the average pore size increases with increasing mixing pressure. As the pressure increases, the outflow from the head is more turbulent, with a higher discharge velocity, which on the one hand causes more energy to be injected into the mixing process, and on the other hand adversely affects the nucleation process, resulting in the formation of larger pores, with a greater number of open pores. The mechanism for the formation of larger pores is proposed in Figure 6.

The average aspect ratio (Figure 7) determined for foams made at different pressures taken from sites A and B is approximately 1.2. The increase in mean pore diameter for higher mixing pressures and the reduction in the number of closed cells is due to the tendency for closed bubbles to burst and for several to merge into one larger bubble, as schematically shown in Figure 8.

Increasing pore diameters and opening up cells adversely affects parameters such as thermal insulation, which depends on the size of the cells [21,22].

The content of closed cells in the foams (Figure 9) tends to decrease with increasing mixing pressure.

The part of the material that forms the structure between the pores adheres to the wall of the resulting large pore and increases according to the scheme, as shown in Figure 8.

Apparent density is the most common and basic parameter that is easy to measure, especially considering industrial production possibilities. It was measured in every block in three different localizations inside the block. For measurements taken from samples from the center of the block, 4–7% of the difference was observed for blocks manufactured with pressures 110, 130 and 150 bars between localizations from the center and sides. The following results are shown in Figure 10 and Table 3.

According to the results of the apparent density test, it is observable that the density of the center of block foam is smaller than the density of foam from the sides of the block.

For block 170, it is noticeable that the results of apparent density in region A diverge from the B region. According to the pore size, which is higher for the center of the foam (Figure 5), the apparent density decreases. Bigger pores are produced because of the higher temperature in the middle of block, where gases locked inside pores expand more than colder gas. The well-known isobaric process confirms this phenomenon with the equation:(2)$$W=\overset{b}{\underset{a}{\int }}p*dV$$
where *W* is work performed by a body of gas on its surroundings, *P* is pressure, *V* is volume, and *a* and *b* are initial and final volumes.

For an ideal gas, a constant value in Equation (3) implies the volume increase for temperature increase.
(3)$$\frac{V}{T}=const$$
where *V* is a volume of gas inside one pore and *T* is the temperature of this gas.

Due to polyurethane synthesis being an exothermic process, the temperature of polyurethane increases and heats the gas inside the pores, causing expansion. The temperature remains longer than the gel time, and lower density is the result in the middle of the block.

For many foam applications, embrittlement is also essential, so for the tested foams, it was also assessed, and its results are summarized in Table 4. The embrittlement of the samples from the outer part of the blocks is lower than for the foams from the central part. For the block made at 130 bar, the embrittlement is lowest for each of the sampling locations. For blocks made at pressures of 130 bar and above, the difference in embrittlement between the sampling locations is above 20%.

The change in water absorption of the produced foams was also analyzed; its results are presented in Figure 11. A tendency to decrease in value was observed with increasing mixing pressure, and it is probably the result of the difference in the chemical structure of these foams, in spite of the fact that the polyurethane system that was used for this experiment has very good parameters of water uptake [23]. The difference in water absorption between the central part of the block and the outer part decreases with increasing mixing pressure.

The compressive strength of samples taken from different locations in the blocks was also characterized and evaluated along and across the growth direction of the foams (Figure 12). As is the case for other groups of rigid foams, the compressive strength of samples compressed along the growth direction is higher than for foams compressed across the growth direction.

The compressive strength does not change significantly for blocks produced at a mixing pressure of 110–150 bar, while it is noticeably lower for a block produced at 170 bar. The change in compressive strength, like that for other rigid foams, varies, as does the apparent density. In the case of the block made at the substrate mixing 170 bar, the pore size of this foam is the largest, which reduces the compressive strength of this foam.

Differences in the strength of the foams depending on the sampling location were analyzed (Table 5). The differences in the compressive strength of the foams depending on the sampling location are within the limits of its determination error.

### 3.3. Effect of Substrate Temperature

Machine manufacturers allow a temperature variation of 1 °C for raw materials. This rule was adhered to in work carried out. The viscosity of the substrate mixture can affect the mixing process, as the viscosity of the polyol is significantly higher than that of the isocyanate, so it was decided to analyze changes in the viscosity of the polyol at the temperatures used in the experiment (Figure 12).

The nature of the changes in the viscosity of the polyol is similar to the nature of the changes in pore wall thickness for the outer part of the blocks (B) (Figure 13). In order to assess what effect differences in the viscosity of the foams had on the chemical structure, an analysis using FTIR infrared spectroscopy was performed. Figure 14 summarizes the FTIR spectra of samples taken from the outer part of the foam blocks (A). The spectrum for sample 150 A shows characteristic bands typical of rigid foams.

Quantitatively, changes in multiple band components of carbonyl groups (Figure 15) were analyzed, and the procedure for multiple band analysis is described in the paper [24]. The multiplet bands were scaled relative to the 1593 cm^−1^ band from the C-C groups in the aromatic groups. Based on the multiple band component analysis results, the proportion of urea groupings (part of urea groups) and of urethane groupings (part of urethane groups) in the hard phase of the tested foams was calculated. The index of hydrogen bonds (IH) connecting the rigid segments in the tested foams was also calculated.

The band from PIR groups at 1412 cm^−1^ (Figure 16) was also quantitatively analyzed based on spectra scaled against the 1593 cm^−1^ band.

The results of the analysis of the proportion of urea and urethane groups and the hydrogen bonding index in the rigid foam segments are summarized in Figure 17 and Figure 18, respectively.

The proportion of urea groupings in part A of the block decreases with increasing mixing pressure in the range of 110–150 bar and then increases at 170 bar. The reaction of NCO groups with water, resulting in the formation of urea bonds, competes in foam synthesis with the reaction of polyol functional groups with NCO groups. Therefore, when the proportion of urea bonds increases, the proportion of urethane bonds in the A part of the block decreases.

In the middle part of block (B), the differences in the proportion of urea and urethane groups in the materials made at different mixing pressures are small, which may be due to slight differences in the temperature of the interior of the block. The changes in the proportion of urea and urethane groups in the outer part of the block (A) with the change in mixing pressure may be a difference in the viscosity of the substrates due to differences in their temperature before the process (Table 1). In the initial stage of the reaction, the higher the viscosity of the substrates, the slower the urea bond formation reaction will occur. However, in the case of a block made at 170 bar, the energy supplied to the substrate mixture causes the reaction to occur much faster, with the result that the growth and gelation time of this foam is significantly reduced (Table 2).

The index of hydrogen bonds connecting the rigid foam segments in samples taken from parts A and B of the foam blocks was also calculated (Figure 18).

With increasing mixing pressure in the range of 110–150 bar, Ih decreases in the outer part of the block (A), after which, at 170 bar, Ih increases. Conversely, in the central part of the blocks (B), Ih increases in the mixing pressure range of 110–150 bar, followed by a decrease at 170 bar. The nature of the changes in Ih in the central part of the blocks varies, as does the gel time of the foams.

The proportion of PIR groups decreases with increasing mixing pressure for samples taken from the outer part of the foam blocks (Figure 19). For samples taken from the central part of the B-blocks, the proportion of PIR decreases when a pressure of 130 bar is applied compared to samples obtained at 110 bar and then increases with increasing mixing pressure.

The effect of viscosity on the PIR contribution and friability of the foams from the central part of the blocks was analyzed (Figure 20). Viscosity was calculated by matching the temperature of the polyol at the time of pouring with the viscosity of the polyol measured at different measurement temperatures according to Brookfield’s method.

Increasing the viscosity of the polyol is associated with an increase in the proportion of PIR groupings in the central part of the block and an increase in brittleness (Figure 20). In the central part of the foam blocks (B), with increasing polyol viscosity, the proportion of PIR increases. The increasing viscosity is the result of a reduction in the temperature of the polyol and, therefore, also of the substrate mixture, with increasing viscosity reducing the mobility of raw materials, yet the proportion of PIR groupings formed increases. The likely reason for the increase in the proportion of PIR as the viscosity of the polyol increases is the longer retention of heat generated by the reaction. With increasing polyol viscosity, the brittleness of the foams also increases. The PIR groupings stiffen the polyurethane macromolecules, resulting in reduced mobility and, consequently, increased brittleness.

## 4. Discussion and Conclusions

The foaming process of polyurethane foams was carried out using various machines, including machines in which mixing was carried out at high pressure. This paper analyzes the effect of mixing pressure in the range of 110–170 bar on the structure and properties of the blocks. This paper analyzed the change in the characteristics of foams from the outer part of the blocks (A) and the middle part (B). As a result of these analyses, it was found that foams made at 170 bar differ in properties from foams made at pressures of 110–150 bar. Foam from a block made at a mixing pressure of 170 bar is characterized by pores of the largest size in the range of 420–490 m, and at other pressures 390–440 m. Foam made at 170 bar contains the fewest closed pores and pores with the largest wall thickness. The apparent density of this foam is the lowest, but its brittleness is the highest, and its compressive strength is the lowest. Differences in properties between the inner and outer parts of the foam blocks were also observed.

In the course of analyzing the foams, it was found that in addition to mixing pressure, the temperature of the substrates also influenced their properties. However, the substrate temperature recommended by machine suppliers was used in the foam synthesis process. The temperature of the substrates affects viscosity during mixing, and this affects the chemical structure of the foam blocks. It was shown that the proportion of urea bonds in the foams, the index of hydrogen bonds in the hard phase of the foams and the proportion of isocyanurate groups changed. In the middle part of the foams, no big differences were observed in the share of urea bonds and the index of hydrogen bonds connecting the hard phase of the foams. There was a significant change in the proportion of isocyanurate groups inside the block, along with a change in the viscosity of the substrates.

The formation of pore nuclei depends on the energy supplied in the mixing process. On the one hand, the pressure should be high enough for bubble release to occur and for proper mixing of the substrates. However, it must not be so large that pore bursting and inhomogeneity of pore size occur, as is the case in foam formed at 170 bar.

Analyzing the properties of blocks shaped in an open mold, significant differences were observed in their properties depending on mixing pressure and substrate temperature, even though the temperature was within the range allowed by machine suppliers. Differences were also observed in the properties of foams taken from different areas of the block. The results of the study indicate that when starting the production of cut products from the block, it is worth analyzing the selection of the parameters of the process of their manufacture in order to obtain products of reproducibly high quality and homogeneity of characteristics.

## Figures and Tables

**Figure 1 materials-16-00857-f001:**
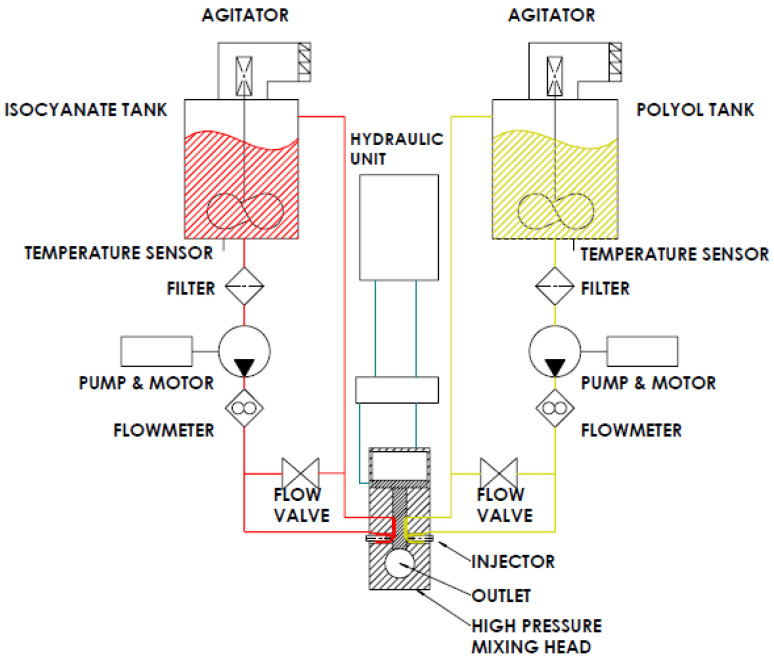
General scheme of a high-pressure machine.

**Figure 2 materials-16-00857-f002:**
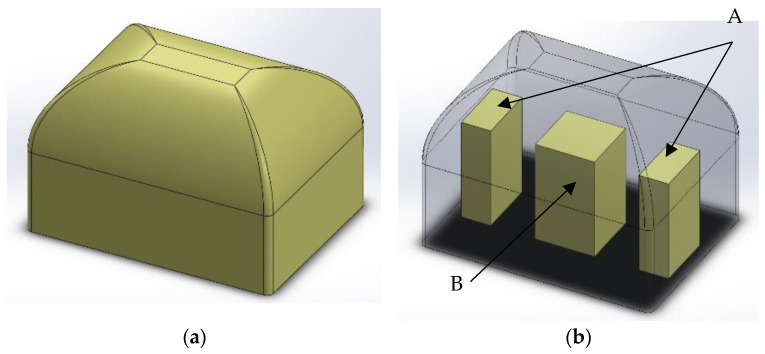
A 3D schematic model of sample location in a foam block. (**a**) Manufactured block; (**b**) localization of specimens.

**Figure 3 materials-16-00857-f003:**
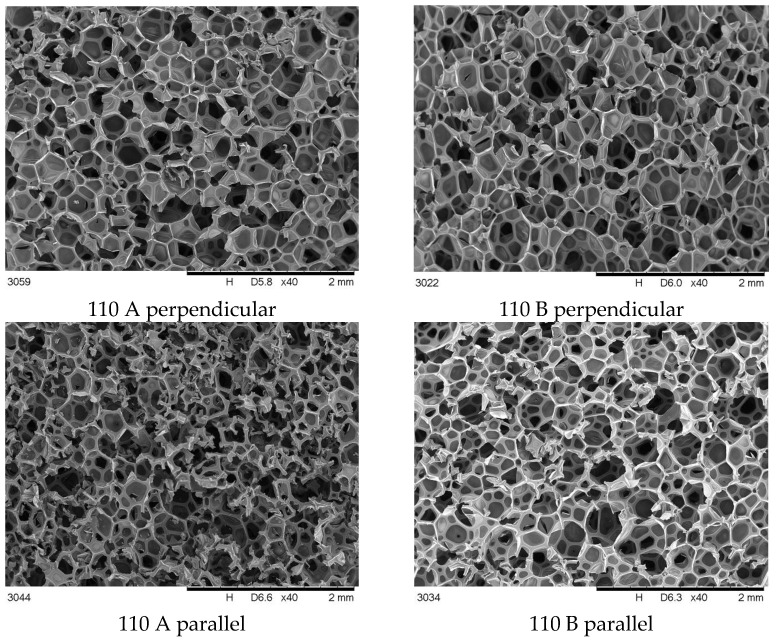
SEM cross-sectional images of foams synthesized at 110 bar, taken perpendicular and parallel to the growth direction.

**Figure 4 materials-16-00857-f004:**
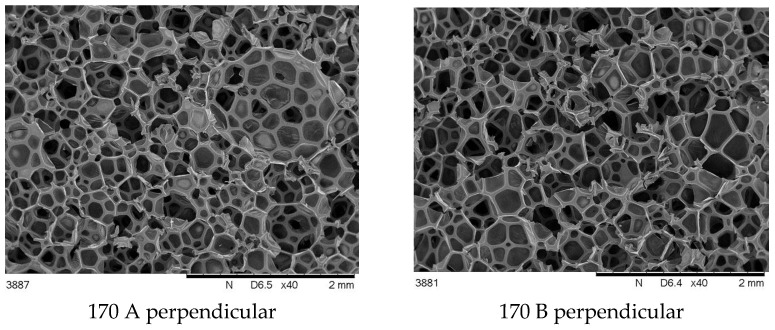
SEM cross-sectional images of foams synthesized at 170 bar, taken perpendicular and parallel to the growth direction.

**Figure 5 materials-16-00857-f005:**
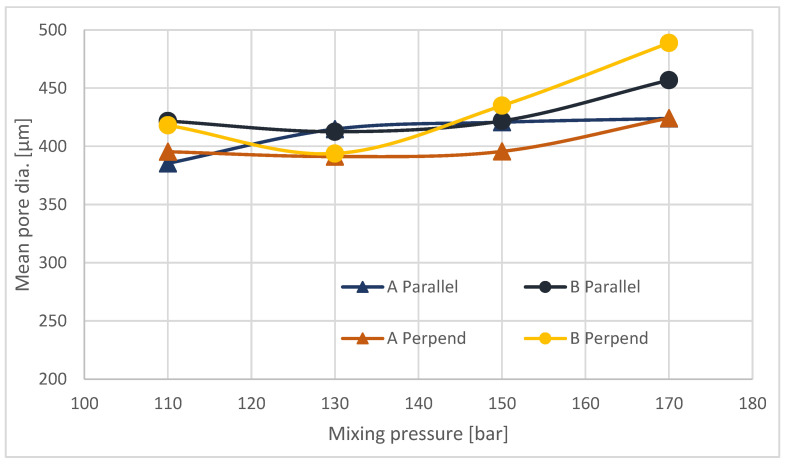
Mean pore diameter for parallel and perpendicular to the growth direction cross sections.

**Figure 6 materials-16-00857-f006:**
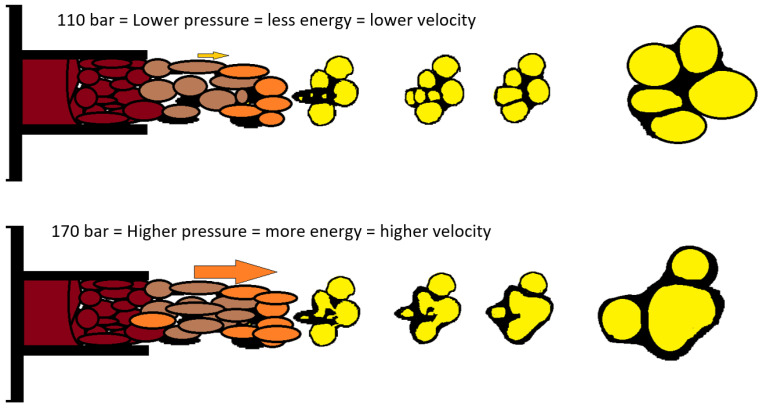
Scheme of injection and pore merging.

**Figure 7 materials-16-00857-f007:**
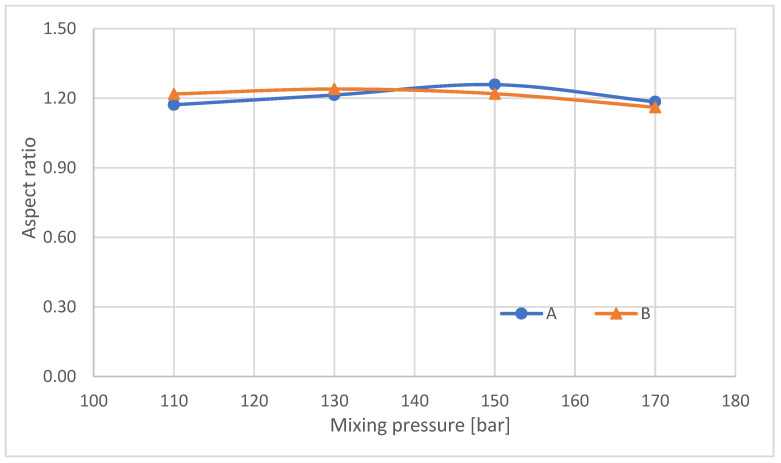
Mean aspect ratio of analyzed RPUFs.

**Figure 8 materials-16-00857-f008:**
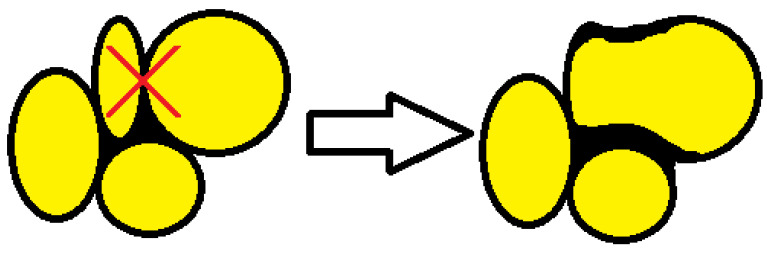
Bubble merging tendency.

**Figure 9 materials-16-00857-f009:**
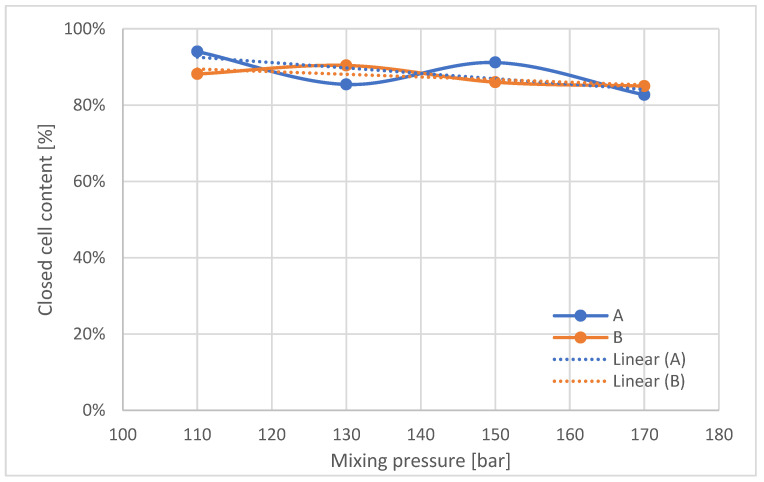
Closed cell content in manufactured polyurethane blocks, broken down by pressure.

**Figure 10 materials-16-00857-f010:**
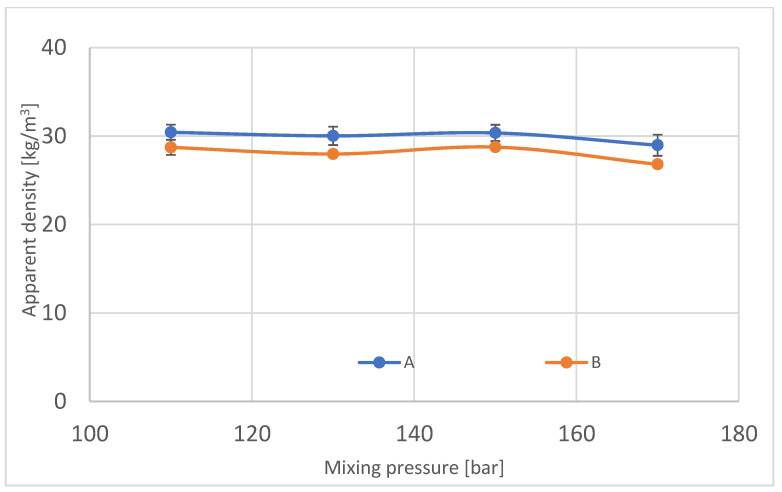
Apparent density in manufactured polyurethane blocks, broken down by pressure.

**Figure 11 materials-16-00857-f011:**
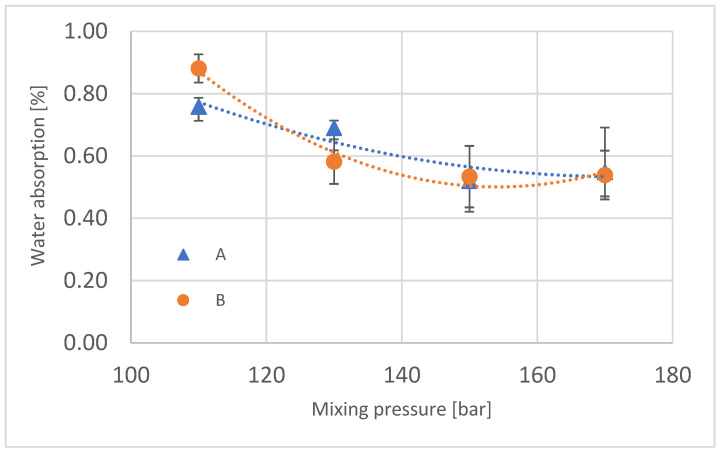
Water absorption of manufactured polyurethane blocks.

**Figure 12 materials-16-00857-f012:**
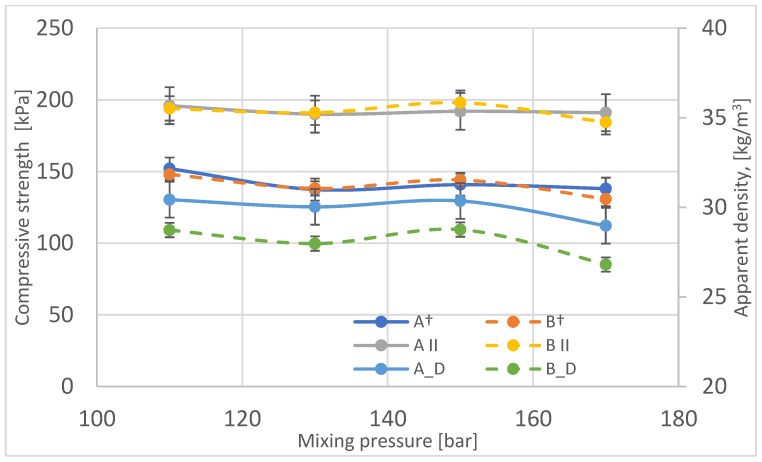
Variation of compressive strength as a function of mixing pressure in the perpendicular (A†, B†) and parallel directions (A ׀׀, B ׀׀) compared to changes in apparent density of foams (A_D, B_D).

**Figure 13 materials-16-00857-f013:**
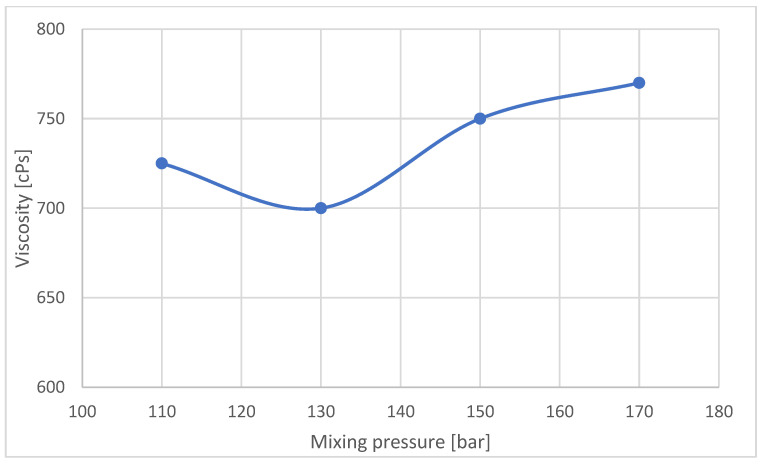
Polyol viscosity according to the mixing pressure during the mixing cycle.

**Figure 14 materials-16-00857-f014:**
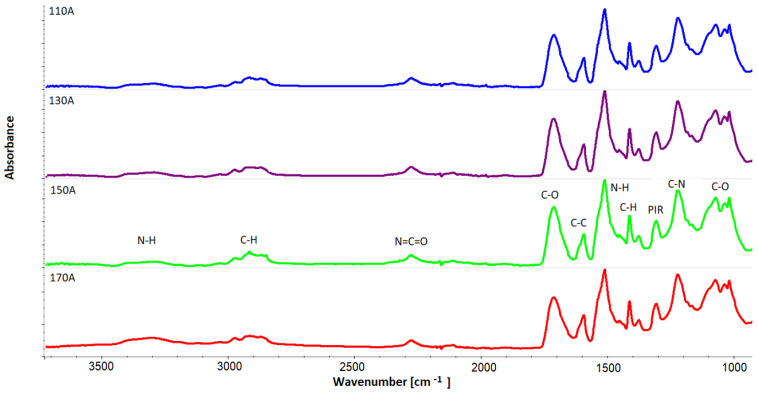
FTIR spectra of samples taken from the outer part of the foam blocks (A).

**Figure 15 materials-16-00857-f015:**
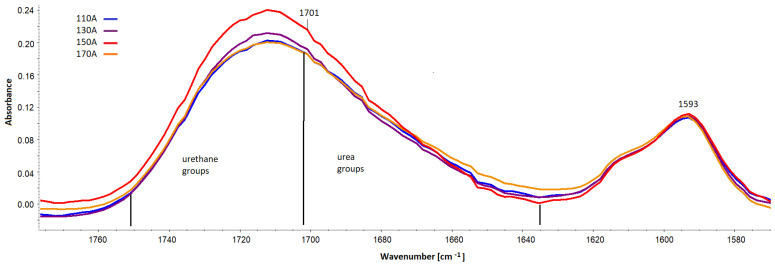
Spectral fragment with a band of carbonyl groups in the range 1635–1750 cm^−1^ and a band serving to scale from the C-C bonds in the aromatic group (1593 cm^−1^).

**Figure 16 materials-16-00857-f016:**
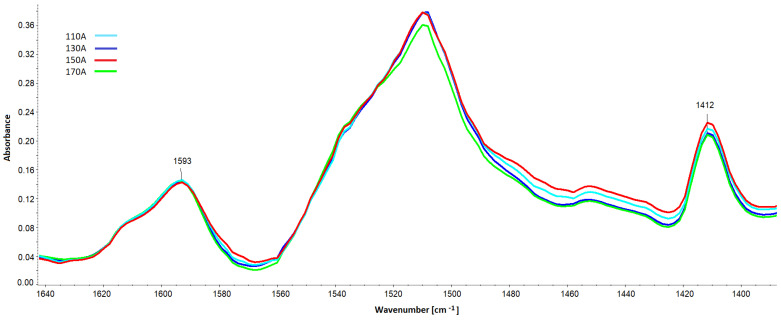
Spectral fragment with the band of polyisocyanurate PIR groups at 1412 cm^−1^ and the band serving to scale from the C-C bonds in the aromatic grouping (1593 cm^−1^).

**Figure 17 materials-16-00857-f017:**
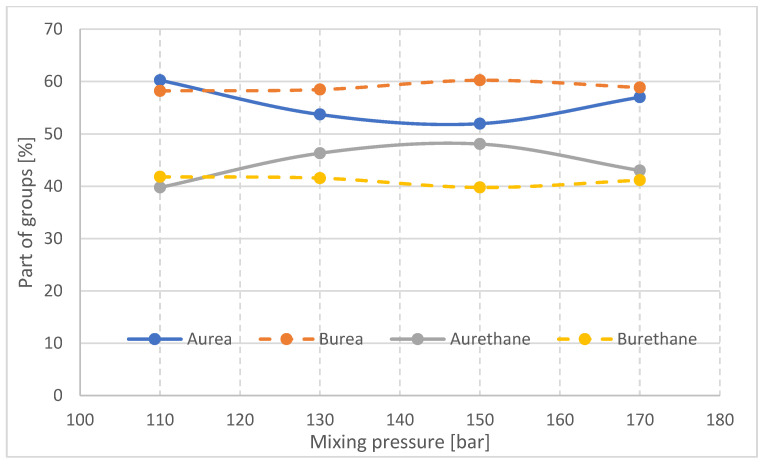
Proportion of urea and urethane groups in rigid foam segments in samples taken from parts A and B of foam blocks.

**Figure 18 materials-16-00857-f018:**
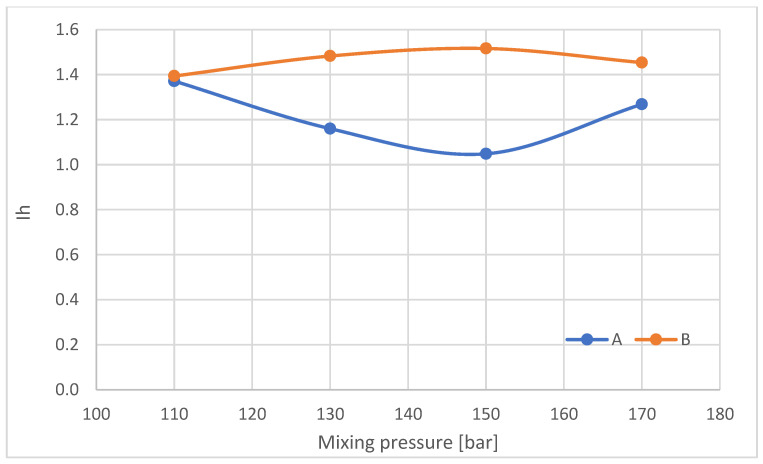
Share of hydrogen bonds connecting the rigid foam segments (Ih).

**Figure 19 materials-16-00857-f019:**
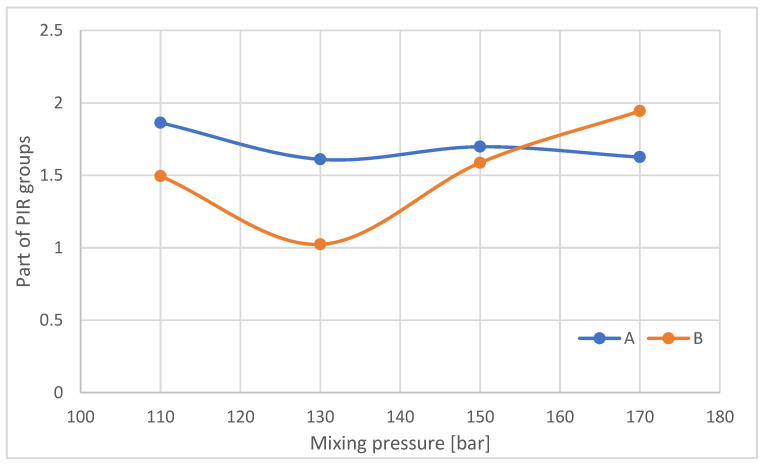
Proportion of PIR groups in samples taken from parts A and B of the foam blocks.

**Figure 20 materials-16-00857-f020:**
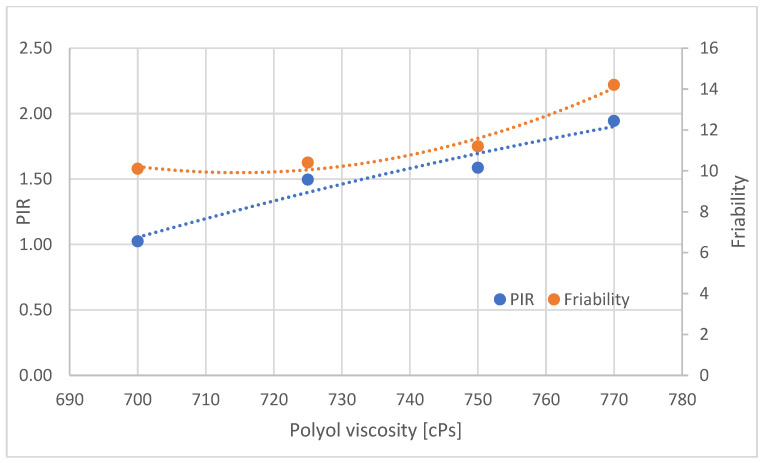
Proportion of PIR and brittleness groups in samples taken from part B of foam blocks.

**Table 1 materials-16-00857-t001:** Synthesis parameters such as component temperatures, ambient temperature and pressure.

Mixing Pressure [bar]	Polyol Temperatures [°C]	Isocyanate Temperature [°C]	Ambient Temperature [°C]	Ambient Pressure [hPa]
110	23.5	22.6	18.3	983
130	23.9	23.0	18.3	983
150	23.2	23.6	18.3	982
170	23.1	23.0	18.4	982

**Table 2 materials-16-00857-t002:** Characteristic times for polyurethane synthesis.

Mixing Pressure [bar]	Start Time [s]	Rise Time [s]	Gel Time [s]	Tack-Free Time [s]
110	10	159	191	270
130	10	161	190	276
150	8	160	189	280
170	7	135	160	260

**Table 3 materials-16-00857-t003:** The differences of apparent density between localization in blocks.

	Mixing Pressure [bar]	110	130	150	170
Localization		[%]	[%]	[%]	[%]
The change in apparent density between A–B [%]		−6	−7	−6	−8

**Table 4 materials-16-00857-t004:** Friability results.

	Mixing Pressure [bar]	110	130	150	170
Localization		[%]	[%]	[%]	[%]
A		9.5	7.9	8.8	10.2
B		10.4	10.1	11.2	14.2
The change in friability between A–B [%]		9	22	21	28

**Table 5 materials-16-00857-t005:** Variation in compressive strength between foam samples taken from different locations.

	_Mixing Pressure [bar]_	110	130	150	170
_The Change in_ _Compressive Strength Between_		[%]	[%]	[%]	[%]
A–B in a perpendicular direction		3	−1	−2	6
A–B in a parallel direction		1	−1	−3	4

## Data Availability

Not applicable.

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
