# Peer review of "The Effect of Mixing Pressure in a High-Pressure Machine on Morphological and Physical Properties of Free-Rising Rigid Polyurethane Foams—A Case Study"

_materials, 2023, doi:10.3390/ma16020857_

Round 1

Reviewer 1 Report

The article analysis mixing pressure impact on the final performance of polyurethane foams. The idea is great, however, the additional information and discussion should be highly improved.

1) In introduction section, authors mention that manufacturers recommend the pressure range between 100 and 200 bar. Why authors did not take into consideration the suggested pressure range and tested polyurethane foams only from 110 to 170 bar?

2) Discussion part does not contain any references to other authors works and comparisons with other authors works. This part need to be highly improved.

3) Fires are missing upper and lower limits of the results obtained. I do not suggest connecting points from average results with lines, because authors did not test what happens between, e.g. 130 and 150 bar.  If tested in this range, results may be different and the connecting line would look differently.

4) Authors write "Increasing pore diameters and opening up cells adversely affects parameters such as thermal insulation...", however, there are no results presented.

5) the formatting of the whole article should be also improved.

6) Please increase the scaling of Figure 14 and Figure 15 values in axes.

7) Introduction section is not of a great quality and reviewer would like to suggest deeper analysis of what has been already done in the same field. Accordingly, the reference list should be broadened.

8) Abstract part is also poorly written, it should contain the most important achievements of the research conducted because this is the first part readers would read in order to decide if the article is worth reading.

9) The materials used for the production of polyurethane foams were not described at all. There is no information about the mixing machine used.

10) If these polyurethane foams are to be used as thermal insulators, why there are no results of thermal conductivity and at least short-term water absorption?

In conclusion, the article needs major revision especially with the regards to discussion of the results presented. 

Author Response

Dear Reviewer, 

I wanted to thank you for your time and effort. I hope the revision that we have prepared is good enough to fit your demands, and soon the article can be accepted. 

In the attachment, I have uploaded some answers to your questions.
Thanks once again.
Kind regards, 
Grzegorz Węgrzyk

Reviewer 2 Report

The paper presents the study on the characterization of polyurethane foam, prepared at various pressures. A lot of results are presented, which are poorly explained - the explanations are in the form of describing how the curves increase or decrease with little or no explanation what the results mean and how they effect the performance or applicability of the foams. If the paper is accepted the following comments should be taken into consideration:

- line 45 - what is the difference between production defects and defects?

- Table 1 - why were the temperatures of the polyol and isocyanate different?

- Table 2 - how did the authors determine the cream time, rise time, gel time, tack-free time? What does it mean that the foam changed the color into creamier? How did the authors quantify the change? 

Figure 4 presents mean pore diameter. How was this parameter determined? How many samples were used for the determination? What was the experimental error?

- line 164 - Figure 7 instead of Figure 5?

- Figure 8 - can we really talk about decreasing of the closed cell content with increasing mixing pressure?

- Line 172 - how was the thickness of the pore walls determined? How many samples were used for the determination? What was the experimental error?

- Lines 178-179 - there is the difference between the determined pore wall thickness and the typical low-density foams? The authors should comment this finding.

- Lines 234 - What was the determination error?

- Table 5 - What does C40 mean?

- Figure 12 should present the viscosity according to the temperature during mixing, but on the abscissa there is mixing pressure? What does it mean? How was the viscosity determined? At different pressures or at different temperature? How was temperature determined?

- Lines 283-287 - the explanation of the obtained results is missing

- Figure 19 - Are the numbers of mixing pressure on abscissa correct?

- Discussion should be more detailed- what are the most important findings and what are the most important scientific contributions of the study?

Author Response

(The authors gave the same response as above.)

Round 2

Reviewer 1 Report

Authors have improved their article according to my comments.

Reviewer 2 Report

English should be improved.